# TEMPORAL DISTRIBUTION-AWARE QUANTIZATION FOR DIFFUSION MODELS

## ABSTRACT

Diffusion models for image generation have achieved notable success in various applications. However, these models often require tremendous storage overhead and inference time cost, severely hampering their deployment on resource-constrained devices. Post-training quantization (PTQ) has recently emerged as a promising way to reduce the model size and the inference latency, by converting the float-point values into lower bit-precision. Nevertheless, most existing PTQ approaches neglect the accumulating quantization errors arising from the substantial distribution variations across distinct layers and blocks at different timesteps, thus suffering a significant accuracy degradation. To address these issues, we propose a novel temporal distribution-aware quantization (DAQ) method for diffusion models. DAQ firstly develops a distribution-aware finetuning framework to dynamically suppress the accumulating quantization errors in the calibration process. Subsequently, DAQ employs a full-precision noise estimation network to optimize the quantized noise estimation network at each sampling timestep, further aligning the quantizers with varying input distributions. We evaluate the proposed method on the widely used public benchmarks for image generation tasks. The experimental results clearly demonstrate that DAQ reaches the state-of-the-art performance compared to existing works. We also display that DAQ can be applied as a plug-and-play module to existing PTQ models, remarkably boosting the overall performance. The source code will be released upon acceptance.

## 1 INTRODUCTION

In recent years, the diffusion model (Ho et al., 2020; Song et al., 2021b;a; Rombach et al., 2022) has become a promising alternative of the conventional generative models including GAN (Goodfellow et al., 2020) and VAE (Kingma & Welling, 2014), due to the high quality and diversity of its generated images, as well as the stable training process. It has a wide range of applications such as the super-resolution (Saharia et al., 2022; Li et al., 2022a; Kadkhodaie & Simoncelli, 2020), graph generation (Niu et al., 2020), image translation (Sasaki et al., 2021), and image restoration (Song et al., 2021b; Kadkhodaie & Simoncelli, 2020). Generally, the generation process of diffusion models involves gradually adding Gaussian noise to image data and then iteratively removing the noise step by step through a noise estimation network. As this process typically takes hundreds or even thousands of steps to find sampling trajectories for denoising, the diffusion model usually requires tremendous storage overhead and inference time cost. For example, the representative Stable Diffusion (Rombach et al., 2022) with DPM-Solver (Lu et al., 2022a) requires 16GB memory and 10GB VRAM during inference, taking seconds to generate a 512×512 resolution image (He et al., 2024b).

The high computational complexity of diffusion models is mainly attributed to the following two reasons. First, generating a single image requires hundreds or even thousands of denoising steps, which involve repeatedly executing the estimation network. Second, the estimation network alone introduces significant computational cost to the generation process (*e.g.* LDM, Stable Diffusion). Despite that many approaches have been proposed to deal with the first issue by reducing the number of estimation steps, balancing the number of steps with the quality of generated images remains a bottleneck. In this paper, we aim to tackle with the second issue, *i.e.* accelerating the UNet-based (Ronneberger et al., 2015) noise estimation network.

Existing works on accelerating the noise estimation network have been focused on quantization (He et al., 2024b; Shang et al., 2023; Li et al., 2023; So et al., 2024; He et al., 2024a; Wang et al., 2024; Huang et al., 2024), distillation Meng et al. (2023); Sun et al. (2023); HUANG et al. (2023), and pruning Li et al. (2022b). Among these techniques, the quantization method has received a lot of attention by converting the weights and activations from floating-point numbers to low-bit-width integers. Typically, quantizing a full-precision model to 8-bit can accelerate the inference process by about 2.2 times (Jacob et al., 2018), with further reduction to 4-bit achieving an additional 59% improvement over the 8-bit setting (He et al., 2024b). However, directly applying the quantization methods designed for general purpose to diffusion models often yields poor performance, as the diffusion models using the same network to denoise inputs with different distributions at various timesteps, which is not handled by the general quantization methods. PTQ4DM (Shang et al., 2023) and Q-Diffusion (Li et al., 2023) attempt solving this problem by incorporating multi-timestep calibration into the quantization process, while other approaches focus on the temporal characteristics within the estimation network to mitigate the impact of multi-timestep distributions (So et al., 2024; Huang et al., 2024). Despite these advancements, a significant performance drop persists when models are quantized to bit-widths lower than 8-bit using post-training techniques. To pinpoint the source of this performance degradation, we analyze quantization errors within the estimation network across different timesteps. Our analysis reveal that the reconstruction granularities employed in quantization are often inappropriate for diffusion models, leading to pronounced discrepancies among quantized modules. Moreover, we identify substantial quantization errors in specific modules characterized by a wide range of activation distributions across timesteps, which contributes to diminished performance when quantizing the estimation network to lower bit-widths.

To address the above issues, we propose a novel method dubbed temporal distribution-aware quantization (DAQ) for diffusion models, to deal with the uneven distribution of internal quantization errors within the network, as well as the accumulation of external quantization errors over multiple sampling timesteps. Unlike previous approaches that focus on calibration components or optimize specific modules separately (Li et al., 2023), we assess the degree of under-recovery in quantized modules by analyzing relative quantization errors and input distributions. This enables our fine-tuning framework to effectively mitigate quantization errors arising from dynamic activation distribution changes within network modules and the significant disparities among different quantized modules. Furthermore, to reduce the accumulation of quantization errors across multiple sampling timesteps in diffusion generative models, we present a parameter finetuning method that suppresses cumulative errors over timesteps. We identify the modules and parameters requiring finetuning based on their degree of under-recovery, using the output of the full-precision model at each sampling timestep as a reference. This method allows for incremental finetuning of quantization factors, thereby reducing cumulative quantization errors.

The main contributions of our work are summarized as follows:

- We propose a novel temporal distribution-aware quantization (DAQ) method for diffusion models, by reducing the accumulating quantization errors arising from the substantial distribution variations across distinct layers and blocks at different timesteps.

- We develop a distribution-aware finetuning framework to dynamically suppress the accumulating quantization errors in the calibration process, and employ a full-precision noise estimation network to optimize the quantized noise estimation network for further aligning the quantizers with varying input distributions. Both of them are plug-and-play modules that are applicable to existing quantization approaches.

- We conduct extensive experiments on various benchmarks, which clearly demonstrate the effectiveness of the proposed method compared to the state-of-the-art approaches.

## 2 RELATED WORK

Existing approaches for accelerating the inference of diffusion models can be roughly divided into two categories. One category of approaches aims to find effective sampling trajectories, either by reducing the number of steps required or by selecting more efficient steps. The other category focuses on minimizing the time and memory overhead for each estimation in the denoising process. In this paper, we introduce specialized quantization methods to enhance the single denoising process.

These methods can be used as plugins to complement other quantization techniques for diffusion models.

## 2.1 Efficient Diffusion Models

In recent years, significant work has focused on accelerating the inference speed of diffusion models, primarily by reducing the number of timesteps required for sampling. Some approaches attempt to transform the diffusion process into a non-Markovian process while keeping the objective function unchanged, thereby eliminating the dependency on chain sampling (Song et al., 2021a). Given that diffusion models use continuous-time sampling, other methods have reformulated the denoising problem into solving differential equations, utilizing differential equation solvers to quickly find approximate solutions (Lu et al., 2022a; Bao et al., 2021; Liu et al., 2022; Lu et al., 2022b).

However, these methods often require the original training data and additional training processes, making them unsuitable for low-resource scenarios. Consequently, some efforts have shifted towards optimizing the denoising network itself by employing techniques such as distillation (Meng et al., 2023; Sun et al., 2023; HUANG et al., 2023), pruning (Li et al., 2022b), and quantization (He et al., 2024b; Shang et al., 2023; Li et al., 2023; So et al., 2024; He et al., 2024a; Wang et al., 2024; Huang et al., 2024) to compress the network. Among these techniques, quantization is the most widely used for optimizing denoising networks.

## 2.2 Model Quantization

Quantization is a widely-used compression method for reducing computational and memory costs. To optimize the inference process across all timesteps, we focus on quantizing the noise estimation model used in diffusion models. We specifically propose methods based on post-training quantization (PTQ) rather than quantization-aware training (QAT) due to PTQ's ease of deployment and widespread adoption. Unlike QAT, which requires retraining the quantized model, PTQ directly quantizes the parameters, making its complexity dependent only on the parameters rather than the original training process. As diffusion models increase in size, the advantages of PTQ-based methods become more pronounced.

PTQ typically compresses the bit-width of weights and activations within the network to reduce memory and computational overhead. Quantization methods generally map data to lower-bit integers, and floating-point operations in a full-precision model are converted into corresponding integer operations, enhancing the inference speed of the quantized model (Krishnamoorthi, 2018).

When using linear mapping to quantize a full-precision floating-point model, the weights and activations are typically quantized into low-bit-width integer representations, denoted as $\bar{W}$. This process can be represented by the following equation:

$$\bar{W} = \text{Clip}\left(\text{Round}\left(\frac{W}{S}\right) + Z, C_{min}, C_{max}\right), \tag{1}$$

where $W$ represents the model parameters, $S$ denotes the scaling factor, $Z$ is the zero point offset, and $C_{min}$ and $C_{max}$ are the lower and upper bounds of the mapping range, also known as the quantization range. $\text{Round}(\cdot)$ and $\text{Clip}(\cdot)$ denote the rounding and clipping operations respectively. A straightforward and effective approach to determining the quantization range and factor is to directly minimize the error between a model's outputs before and after quantization. Previous study (Nahshan et al., 2021) has evaluated this using metrics such as L1 distance, cosine similarity, KL divergence, and MSE, ultimately finding that the Lp norm (with p = 2.4) yields the best results (Shang et al., 2023). Additionally, AdaRound (Nagel et al., 2020) introduced an adaptive method for determining rounding directions, which maintains high accuracy even at 4-bit precision. However, when applying PTQ, a small subset of data is still required as a calibration dataset to adjust the network's activations. Consequently, much of the research on PTQ methods has concentrated on optimizing the calibration process.

Several studies have investigated the impact of calibration dataset size on quantization performance. EasyQuant (Wu et al., 2020), for example, directly uses the training data to establish the upper and lower bounds of the quantization range. ZeroQ (Cai et al., 2020) eliminates the need for original training data by generating a calibration dataset from the model's gradient information and utilizes

mixed-precision quantization to determine the optimal bit-width. BRECQ (Li et al., 2021) examines the trade-offs between layer-wise, block-wise, and whole-network calibration, concluding that block reconstruction offers the most effective granularity. In this paper, we explore our framework based on BRECQ. However, traditional PTQ methods do not perform well on diffusion models when we directly apply them.

When quantizing diffusion models, the primary source of inference overhead is the noise estimation network. Existing research has focused on quantizing this network due to its high resource demands (Shang et al., 2023; Li et al., 2023). Given the significant costs associated with training these models, PTQ methods are preferred. These methods require fewer resources, are highly portable, and offer rapid quantization speeds.

In current studies, PTQ4DM (Shang et al., 2023) and Q-Diffusion (Li et al., 2023) analyze the distribution of calibration datasets, suggesting that uniformly sampling images from different timesteps to form the calibration dataset for the noise estimation network can reduce quantization error. Q-Diffusion also proposes an optimization strategy for UNet-based noise estimation networks. They discovered that the residual networks used in UNet (Ronneberger et al., 2015) can amplify quantization errors during skip connections due to varying levels of quantization recovery. To address this, they designed a channel-separated quantization scheme for residual blocks (Li et al., 2023), achieving results comparable to full-precision models on W8A8 and W4A8 on the CIFAR-10 and LSUN datasets. PTQD (He et al., 2024b) found that quantization errors in diffusion models contain Gaussian noise. Considering that gaussian noise is inherent in diffusion models, they merged these noise components and adjusted the variance of the Gaussian noise to reduce quantization errors. They also used the signal-to-noise ratio to evaluate quantization effects and determine the optimal quantization bit-widths at different timesteps, experimenting with mixed-precision quantization strategies on the ImageNet and LSUN datasets. TDQ (So et al., 2024) employs a three-layer perceptron to map sampling time encoding information to finetune parameters for correcting quantization factors. EfficientDM (He et al., 2024a) adopts the QLoRA (Dettmers et al., 2024), directly finetuning quantization factors during the quantization calibration process. TFMQ-DM (Huang et al., 2024) constructs timestep-specific quantization modules to correct errors generated when embedding quantized time information encoding modules. APQ-DM (Wang et al., 2024) groups different timesteps and sets shared parameters across these groups to find suitable mapping ranges for quantization factors.

However, several issues remain in quantizing diffusion models. These include the use of uniform quantization settings across different network modules, insufficient consideration of the distribution characteristics of activation values over time, and the accumulation of quantization errors over multiple timesteps.

## 3 THE PROPOSED APPROACH

In this section, we propose a dynamic finetuning method that is aware of activation distribution ranges to adapt to the multi-sampling time features of diffusion generative models. Furthermore, we suppress cumulative errors during inference by post-processing the quantized models. We begin by identifying the problems in existing quantization algorithms, then proceed with a detailed analysis, and finally, we present our proposed solution.

### 3.1 SIGNIFICANT DIFFERENCE OF DISTRIBUTION BETWEEN DIFFERENT MODULES

#### 3.1.1 DISTRIBUTION DIFFERENCE OF ACTIVATION

When quantizing the noise estimation network of DDIM (Song et al., 2021a), we observe that the outputs of different modules deviate to varying degrees from those in full-precision diffusion models within a single sampling timestep. Additionally, there are notable differences in activation value ranges and quantization errors across different quantized modules. This issue can be attributed to two main factors. Firstly, the post-training quantization methods based on BRECQ (Li et al., 2021) employ globally uniform quantization hyperparameters for modules with varying reconstruction granularities within the network. Secondly, the calibration data samples used during post-training quantization are typically insufficient, usually ranging from 200 to 5000 samples. This is significantly smaller than the size of dataset used for training the pre-trained model, which may lead to both overfitting and underfitting in different modules within the same quantized network.

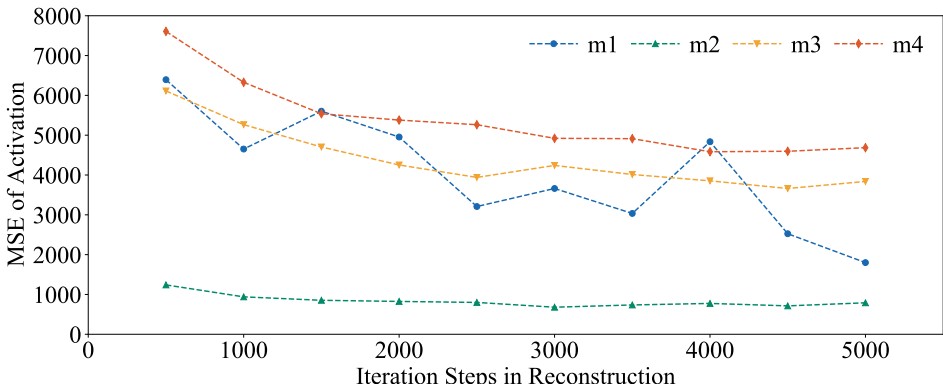

Figure 1: There are significant different behaviors between different modules in DDIM when applying the same quantization settings. Some modules need only 1000 iters while others might need 5000 iters or even more to get the quantization factors.

To validate this hypothesis, we analyze the quantization reconstruction of activation quantizers in different modules across the entire network during the calibrating process of quantization factors. The results in Figure 1 indicate that the number of iterations required for quantization reconstruction varies among different modules, suggesting that the network may exhibit both overfitting and underfitting of activation quantizers when using a unified set of quantization reconstruction hyperparameters. To further illustrate this issue, we conduct experiments using different quantization reconstruction hyperparameters based on BRECQ and identified underfitted activation quantizers in the network. As shown in Figure 2 we find that their quantization loss could potentially be reduced by up to approximately 50%. In response, we set an error threshold to dynamically assess the quantization reconstruction error, thereby mitigating the effects of overfitting and underfitting.

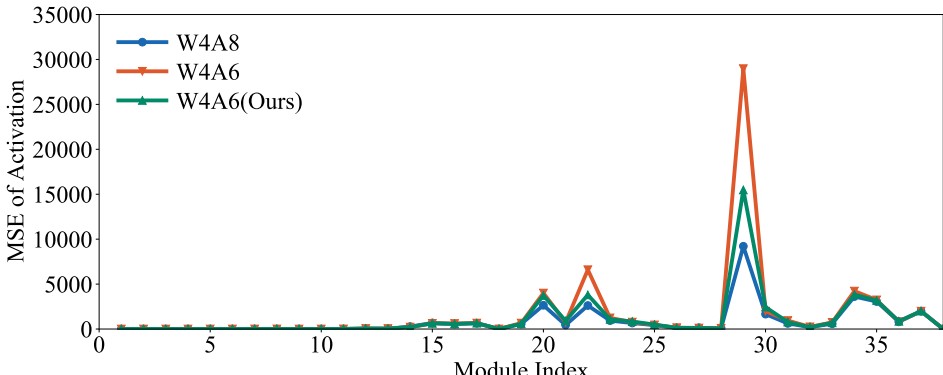

Figure 2: We evaluated the method proposed by calculating mean squared error of activation in each quantized modules. It shows that the bit width of activation affects the error obviously while our framework could reduce it by up to 50%.

Moreover, the quantizers within the network are expected to be adapted to multiple sampling timesteps, requiring a single quantizer in a quantized module to accommodate a range of distributions across different timesteps. However, the noise estimation network with shared parameters needs to run $n$ times for $n$ timesteps, with each timestep using image data whose distribution varies with the sampling timestep as input. By comparing the activation value ranges in different modules at various sampling timesteps, we observe that the same quantizer within a reconstruction module generates activation values with significantly different distribution ranges when processing inputs from different sampling timesteps. This implies that using the same quantization settings for all timesteps is inappropriate, as different timesteps result in different activation distributions. There-

fore, designing a quantization strategy that can dynamically perceive the distribution range of activation values at different sampling timesteps is crucial for effectively quantizing diffusion models.

### 3.1.2 DISTRIBUTION DIFFERENCE BETWEEN TIMESTEPS

In the noise estimation network, the input at each sampling timestep depends on the output from the previous timestep. As a result, the quantization errors introduced by the quantized noise estimation network accumulate over the sampling process, gradually diverging from the outputs of the full-precision network. By comparing the outputs of the full-precision noise estimation network with those of the quantized network at each sampling timestep, we observe that these quantization errors indeed accumulate over time. For instance, the quantization error increases progressively and reaches saturation after 40-50 timesteps when we quantize the DDIM to W4A8 with 100 sampling timesteps. This observation highlights that addressing the accumulation of quantization errors over sampling timesteps is a critical area of focus in the research of quantization methods for diffusion models.

### 3.2 FINETUNING THE DIFFUSION MODELS

#### 3.2.1 DISTRIBUTION-AWARED DYNAMIC FINETUNING FRAMEWORK FOR QUANTIZERS

As mentioned above, the activation distributions differ significantly across quantized modules, often resulting in imbalanced quantization errors within a single timestep. This imbalance can degrade the performance of lower-bit quantization for diffusion models, as it becomes challenging to adequately represent the activation distribution range with low-bit quantizers. To address this issue, we introduce distribution-aware finetuning parameters for the activation quantizers. We use calibration data to finetune the quantization factors for each activation quantizer, similar to the reconstruction process. Specifically, these finetuning parameters are integrated into the quantization factors and subsequently removed, ensuring no additional storage or computational costs are incurred.

For the issue of varying convergence degrees in the quantization reconstruction process across network modules, we hypothesize that the input distribution range ($X_{range}$), relative quantization error ($Q_{err}$), and the degree of under-recovery ($\eta$) are positively correlated. A broader input distribution range can lead to larger clipping errors during quantization, making these clipping errors a more significant component of the total quantization error than rounding errors. However, the error value alone is not the only factor to consider, as it is also related to the reconstruction granularity. This means that some modules with smaller input distribution ranges can still produce significant relative quantization errors. To address this, we evaluated the degree of under-recovery ($\eta$) using the distribution range and relative quantization error during the quantization reconstruction. We then inserted finetuning parameters into modules that either had the top $\beta\%$ of $X_{range}$ or relative quantization errors greater than $\gamma\%$. This approach was intended to accelerate the convergence of quantization factors within these activation quantizers. In this paper, $\beta$ was set to 10 and $\gamma$ to 20.

$$\eta \propto X_{range}, Q_{err}. \tag{2}$$

During the quantization process, we first scale the full-precision data $X_{fp}$ using the scaling factor $S$. Rounding errors are introduced through the rounding operation. After adding the offset $Z$, clipping errors are introduced using the clip operation. The quantization result $X_{quant}$ is obtained after dequantization. Subsequently, using the quantization factors $S$ and $Z$, the simulated quantized data $X_{quant}$ can be restored to full-precision data $X_{dequant}$, replacing $X_{fp}$ and passing it to subsequent modules.

$$X_{quant} = \text{Clip}\left(\text{Round}\left(\frac{X_{fp}}{S}\right) + Z\right), \tag{3}$$

$$X_{dequant} = (X_{quant} - Z) \cdot S. \tag{4}$$

When inserting finetuning parameters $F_S$ and $F_Z$ to finetune the quantization factors, the quantization formula would be transformed into:

$$F(X_{int}) = \text{Round}\left(\frac{X_{fp}}{S \cdot F_S}\right) + (Z + \text{Round}(F_Z)), \tag{5}$$

$$F(X_{quant}) = \text{Clip}(F(X_{int}), C_{min}, C_{max}), \tag{6}$$

$$F(X_{dequant}) = (F(X_{quant}) - \bar{Z}) \cdot \bar{S}. \tag{7}$$

In particular, when the finetuning parameters $F_S = 1$ and $F_Z = 0$, the function $F(X_{dequant})$ yields $X_{dequant}$. Thus, this finetuning method initializes with $F_S = 1$ and $F_Z = 0$ to ensure that the finetuned quantized model does not deviate significantly from the original quantized model. During inference, the finetuning parameters can be merged into the quantization factors in the basic quantization framework, modifying the original quantization factors $S$ and $Z$ to $\bar{S}$ and $\bar{Z}$.

As the input distribution ranges vary across multiple sampling timesteps, traditional quantization methods often result in significant errors when using a single set of quantization factors for activation value distributions across different timesteps. To address this, we introduced temporal finetuning parameters $F_t$ to dynamically adjust the scaling range. The adjusted quantization process can be expressed as:

$$F(X_{int}) = \text{Round}\left(\text{Round}\left(\frac{X * fp}{S \cdot F_S}\right) \cdot F_t\right) + (Z + F_Z). \tag{8}$$

This adjustment allows the quantization factors to better align with the varying activation distributions over different sampling timesteps.

Experimental results show that the introduction of $F_t$ outperformed the sole inclusion of $F_S$ and $F_Z$, resulting in better IS and FID metrics. Additionally, the activation quantization errors of W4A6 with finetuning are significantly lower than those of W4A6 without finetuning, and closer to the performance of W4A8. This performance improvement aligns with the use of a distribution range-aware dynamic finetuning framework for quantization factors under the W4A6 quantization bit width.

In summary, when finetuning quantization factors using the proposed method, we consider the sampling timesteps, the activation distribution range ($X_{range}$) within a single quantizer, and the original quantization factors ($S$ and $Z$). The decision to finetune is based on the degree of under-recovery ($\eta$) of the quantizer. Finally, the original quantization factors are integrated with the finetuning parameters ($F_S$ and $F_Z$) during the inference stage.

### 3.2.2 TIMEWISE FINETUNE FOR SELECTIVE QUANTIZATION PARAMETERS

As diffusion models estimate noise multiple times using the same network, quantized models often accumulate quantization errors over multiple sampling timesteps, leading to significant deviations. To address this issue, we design a post-processing method that operates at each timestep to reduce the impact of these errors. We use the full-precision network as a reference for the quantized network to minimize the difference between the quantized model and the original model. At each timestep, we compare the output of the quantized model with that of the full-precision model to calculate the error. This helps the quantized modules achieve performance closer to that of the full-precision network. By guiding the quantized model to suppress cumulative quantization errors at each step, we can improve overall accuracy and stability.

Also, given the large scale of parameters in diffusion generative models, finetuning all parameters is prohibitively time-consuming and computationally expensive. Therefore, we selectively finetune specific quantizer parameters in certain modules to reconstruct the quantization error at the network level based on the degree of under-recovery ($\eta$). This approach effectively suppresses the cumulative quantization error over multiple sampling timesteps.

## 4 EXPERIMENTAL RESULTS AND ANALYSIS

### 4.1 IMPLEMENTATION DETAILS

#### 4.1.1 MODELS AND METRICS

To validate the effectiveness of our method, we quantize two widely adopted network architectures: DDIM (Song et al., 2021a) and LDM (Rombach et al., 2022). For the DDIM experiments, we use the CIFAR-10 (Krizhevsky, 2009) dataset. For LDM, we conduct experiments using ImageNet (Deng et al., 2009) and LSUN (Yu et al., 2015). We assess the performance of the diffusion models using

Inception Score (IS) (Salimans et al., 2016) and Frechet Inception Distance (FID) (Heusel et al., 2017). The results are obtained by sampling 50,000 images and evaluating them with both ADM's TensorFlow evaluation suite and torch-fidelity. All experiments are conducted using an RTX 3090 GPU and implemented with the PyTorch framework.

### 4.1.2 QUANTIZATION SETTINGS

When quantizing the noise estimation network, we utilize the AdaRound quantizer (Nagel et al., 2020) for the weights and the uniform quantizer for the activations. For calibrating the quantization factors, we employ 5120 images uniformly sampled from 20 timesteps (256 images per timestep) as calibration data, with a batch size of 32 during calibration. For quantization reconstruction, we implement a post-training quantization framework based on BRECQ (Li et al., 2021). Residual modules and attention modules in the network are reconstructed at the block granularity, while other parts are reconstructed at the layer granularity.

| Methods | Bits(W/A) | CIFAR-10 | | ImageNet | |
|---|---|---|---|---|---|
| | | IS↑ | FID↓ | IS↑ | FID↓ |
| Full Prec. | W32A32 | 9.12 | 4.22 | 235.64 | 10.91 |
| PTQ4DM | W8A8 | 9.31 | 14.18 | 161.75 | 12.59 |
| Q-Diffusion | W8A8 | 9.48 | 3.75 | 187.65 | 12.80 |
| PTQD | W8A8 | - | - | 153.92 | 11.94 |
| TDQ | W8A8 | 8.85 | 5.99 | - | - |
| APQ-DM | W8A8 | 9.07 | 4.24 | 179.13 | 11.58 |
| TFMQ-DM | W8A8 | 9.07 | 4.24 | 198.86 | 10.79 |
| Ours | W8A8 | 9.67 | 3.38 | 216.04 | 12.05 |
| Q-Diffusion | W4A8 | 9.12 | 4.93 | 212.51 | 10.68 |
| PTQD | W4A8 | - | - | 214.73 | 10.40 |
| TFMQ-DM | W4A8 | 9.13 | 4.78 | 221.82 | 10.29 |
| Ours | W4A8 | 9.49 | 4.08 | 213.44 | 10.23 |
| PTQ4DM | W6A6 | - | - | 140.86 | 13.68 |
| Q-Diffusion | W6A6 | 8.76 | 9.19 | 146.41 | 13.94 |
| APQ-DM | W6A6 | 9.06 | 6.57 | 178.64 | 11.58 |
| TFMQ-DM | W6A6 | 8.84 | 9.59 | - | - |
| Ours | W6A6 | 9.40 | 4.61 | 218.28 | 10.67 |

Table 1: Quantization results for unconditional image generation with DDIM on CIFAR-10 $32 \times 32$ and conditional image generation with LDM-4 on ImageNet $256 \times 256$.

### 4.2 MAIN RESULTS

In this section, we compare out proposed method with the state-of-the-art post-training quantization methods including PTQ4DM (Shang et al., 2023), Q-Diffusion (Li et al., 2023), PTQD (He et al., 2024b), APQ-DM (Wang et al., 2024) and TFMQ-DM (Huang et al., 2024). The IS and FID scores of these frameworks are acquired by their released results or our implementation according to the officially released code.

Unconditional generation involves sampling a random variable in diffusion models to produce images with distributions similar to those in the training datasets. We evaluate our post-training quantization methods on the CIFAR-10 ($32 \times 32$), LSUN-Church-Outdoor ($256 \times 256$), and LSUN-Bedroom ($256 \times 256$) datasets (Yu et al., 2015). The quality of image generation is presented in Tables 1 and 2, respectively.

While PTQ4DM and Q-Diffusion introduce methods to form the calibration dataset (Shang et al., 2023; Li et al., 2023), TFMQ-DM focuses on timestep-specific quantization modules (Huang et al., 2024). APQ-DM groups different timesteps and sets shared parameters across these groups to determine appropriate mapping ranges for quantization factors (Wang et al., 2024). However, these methods overlook the quantization differences between various modules within the estimation network.

| Methods | Bits(W/A) | Churches FID↓ | Bedrooms FID↓ |
|---------|-----------|---------------|---------------|
| Full Prec. | W32A32 | 4.12 | 2.98 |
| PTQ4DM | W8A8 | 4.80 | 4.75 |
| Q-Diffusion | W8A8 | 4.41 | 4.51 |
| PTQD | W8A8 | 4.89 | 3.75 |
| APQ-DM | W8A8 | 4.02 | 3.88 |
| TFMQ-DM | W8A8 | 4.01 | 3.14 |
| Ours | W8A8 | 3.68 | 3.73 |
| PTQ4DM | W4A8 | 4.97 | 20.72 |
| Q-Diffusion | W4A8 | 4.66 | 6.40 |
| PTQD | W4A8 | 5.10 | 5.94 |
| TFMQ-DM | W4A8 | 4.14 | 3.68 |
| Ours | W4A8 | 4.17 | 3.71 |
| PTQ4DM | W6A6 | 11.05 | 11.10 |
| Q-Diffusion | W6A6 | 10.90 | 10.10 |
| APQ-DM | W6A6 | 6.90 | 9.88 |
| Ours | W6A6 | 8.41 | 9.04 |

Table 2: Quantization results for unconditional image generation with LDM-8 on LSUN-Churches $256 \times 256$ and LDM-4 on LSUN-Bedrooms $256 \times 256$.

As a result, our method surpasses the state-of-the-art results, achieving an improvement of 0.34 (9.40 vs. 9.06) in IS and 1.94 (4.61 vs. 6.57) in FID on the CIFAR-10 dataset. Also, Table 1 shows the quantization results on the ImageNet $256 \times 256$ dataset. We employ a denoising process with 20 iterations, setting eta and cfg to 0.0 and 3.0 respectively. Compared to APQ-DM (Wang et al., 2024), our method achieves a FID reduction of 0.91 on W6A6. The computational cost remains consistent with baseline methods. In this paper, we implement our method based on Q-Diffusion, and the results showed in Table 1, 2 demonstrate improved performance across all datasets and models. Additionally, our method can be implemented as a plugin for other quantization methods for diffusion models, as it reduces quantization error from a different perspective compared to existing methods.

| Methods | Bits(W/A) | CIFAR-10 IS↑ | FID↓ |
|---------|-----------|--------------|------|
| Full Prec. | W32A32 | 9.12 | 4.22 |
| TDQ | W8A8 | 9.58 | 3.77 |
| TDQ + Ours | W8A8 | 9.58 | 3.47 |
| EfficientDM | W4A8 | 9.30 | 4.67 |
| EfficientDM + Ours | W4A8 | 9.43 | 4.18 |
| Q-Diffusion | W4A8 | 9.12 | 4.93 |
| Q-Diffusion + Ours | W4A8 | 9.43 | 4.02 |

Table 3: Quantization results with DDIM on CIFAR-10 32×32, + Ours represents the application of our proposed method.

To further verify that the method proposed in this paper can be applied as a plugin to other quantization methods, we reproduce results from TDQ (So et al., 2024), EfficientDM (He et al., 2024a), and Q-Diffusion (Li et al., 2023), and then apply our method on top of them. The experimental results in Table 3 confirm that our optimization direction for quantizing diffusion models is indeed orthogonal to other methods.

## 4.3 ABLATION STUDY

In order to demonstrate the influence of the distribution-aware dynamic finetuning framework for quantization factors and the cumulative error suppression method, we conduct the ablation experiments on the DDIM (Song et al., 2021a). And the results in Table 4 show that both the distribution-aware dynamic finetuning framework for quantization factors and the temporal finetuning method

| Methods | Bits(W/A) | CIFAR-10 | |
|---|---|---|---|
| | | IS↑ | FID↓ |
| Full Prec. | W32A32 | 9.12 | 4.22 |
| Baseline | W8A8 | 9.38 | 3.75 |
| +DA | W8A8 | 9.48 | 3.85 |
| +PT | W8A8 | 9.46 | 3.72 |
| +DA +PT | W8A8 | 9.67 | 3.38 |
| Baseline | W6A6 | 8.76 | 9.19 |
| +DA | W6A6 | 8.96 | 6.75 |
| +PT | W6A6 | 9.31 | 8.10 |
| +Ours(+DA +PT) | W6A6 | 9.40 | 4.61 |

Table 4: The effect of different methods proposed in the paper. The experiment is conducted over DDIM on CIFAR-10 $32 \times 32$.

outperform the baseline methods on W8A8, W6A6, and W4A8. Here, +DA (Distribution Aware) indicates the application of the distribution-aware dynamic finetuning framework for quantization factors, and +PT (Parameters Finetuning) indicates the application of the temporal parameters fine-tuning method for cumulative error suppression.

When we quantize DDIM to W8A8, both the IS and FID metrics improve after applying the DA and PT methods, although the FID metrics showed no significant difference compared to the baseline methods. However, when both methods are combined, the performance shows significant improvement. This enhancement can be attributed to the fact that the original quantization factors of the quantizer have limited representational capacity and face performance bottlenecks. Our methods help the original quantizer escape local optima and find better quantization parameters through multi-dimensional optimization. Moreover, using the DA and PT methods individually achieved better results in IS and FID metrics compared to the baseline methods. The combined use of both methods also resulted in better model performance when quantizing diffusion models to W8A8 and W6A6.

## 5 CONCLUSION

This research investigates quantization methods for diffusion generative models. By considering the activation distribution of noise estimation networks and addressing imbalanced quantization across different modules, we have enhanced existing post-training quantization techniques. Our improvements consistently surpass the best available post-training quantization compression methods at the same bit-width. Furthermore, our method is orthogonal to other approaches, making it suitable as a plugin for existing quantization techniques. However, this work has some limitations. The granularity division strategy for quantization reconstruction, adopted from BRECQ (Li et al., 2021) and Q-Diffusion (Li et al., 2023), results in an uneven distribution of quantization errors within the network. This indicates the need for a more detailed examination of internal quantization granularity.

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

# A APPENDIX

## A.1 ACTIVATION RANGES ACROSS MODULES

We find that activation value ranges vary significantly across different modules, with pronounced quantization errors observed in the ResBlock and Downsample layers, as shown in Figure 3. This observation suggests that a distribution-aware approach is crucial for quantizing diffusion models, as it allows for targeted reduction of quantization errors across different modules.

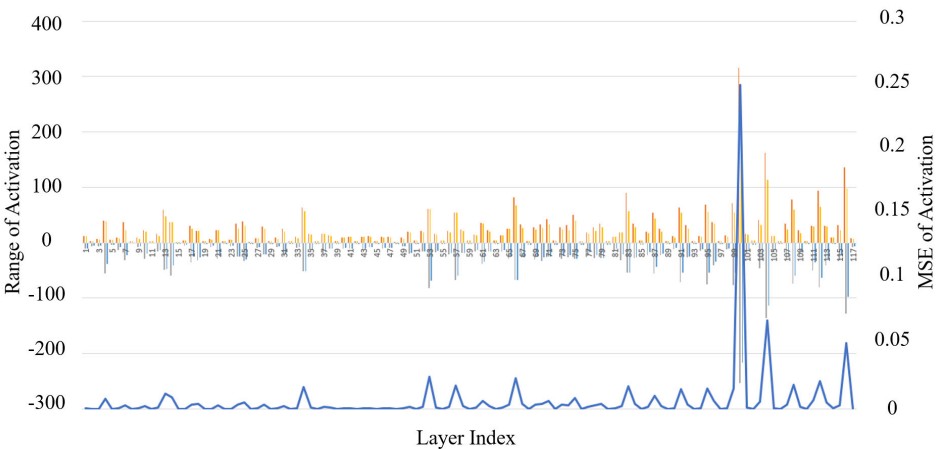

Figure 3: Activation ranges across different modules for DDIM on CIFAR-10 $32 \times 32$ with 100 denoising steps are measured. Also, We calculate the MSE between the full-precision DDIM and the quantized model.

## A.2 MORE DETAILS OF EXPERIMENTS

We sample the calibration and reconstruct modules following the settings of Q-Diffusion with an RTX 3090 GPU, and then apply our finetuning methods after the reconstruction process in each module. However, there are notable differences between various quantization frameworks for diffusion models. We reproduce other methods using either their released code or the algorithms described in their publication. Additionally, the IS and FID scores are measured differently across these methods, which may lead to significant variations in performance compared to the results they reported. We reproduce the SOTA TFMQ-DM method on CIFAR-10 and implement our method on it. As shown in Table 5, we achieve better IS and FID scores on W8A8, W4A8, and W6A6. These results indicate that our approach can be effectively extended to other quantization frameworks.

## A.3 RANGES OF THE HYPERPARAMETERS

In this section, we further explore the ranges of several hyperparameters in Table 6, including calibration size and iterations of reconstructions. The results show that using 256 images for calibration is sufficient, as larger calibration sizes do not improve the performance of the quantized models.

| Methods | Bits(W/A) | IS↑ | FID↓ |
|---|---|---|---|
| Full Prec. | W32A32 | 9.12 | 4.22 |
| TFMQ-DM | W8A8 | 9.07 | 4.28 |
| TFMQ-DM + Ours | W8A8 | 9.19 | 4.12 |
| TFMQ-DM | W4A8 | 9.03 | 7.68 |
| TFMQ-DM + Ours | W4A8 | 9.09 | 6.79 |
| TFMQ-DM | W6A6 | 8.84 | 9.59 |
| TFMQ-DM + Ours | W6A6 | 8.96 | 7.82 |

Table 5: We reproduce TFMQ-DM for DDIM on CIFAR-10 32 × 32 according to the released code and apply our method to their framework.

Additionally, performing more than 40k reconstruction iterations has a noticeably negative impact on FID scores. Consequently, we selected the hyperparameters that provided the best performance within the tested ranges.

| Bits(W/A) | Images in Calibration | IS↑ | FID↓ | Iterations | IS↑ | FID↓ |
|---|---|---|---|---|---|---|
| W32A32 | - | 9.12 | 4.22 | - | 9.12 | 4.22 |
| W4A8 | 64 | 8.57 | 4.89 | 20k | 9.28 | 4.76 |
| W4A8 | 128 | 8.95 | 4.61 | 40k | 9.43 | 4.02 |
| W4A8 | 256 | 9.19 | 4.27 | 60k | 9.18 | 5.65 |
| W4A8 | 512 | 9.04 | 4.47 | 80k | 9.03 | 5.87 |
| W4A8 | 1024 | 9.09 | 4.59 | 100k | 9.14 | 5.41 |

Table 6: Quantization results for DDIM on CIFAR-10 32 × 32 with different calibration size and different reconstruction iterations.. Each image in the calibration will be sampled in 20 timesteps, which means the size of calibration is 20 × the number of images.

## A.4 VISUALIZATION RESULTS

In this section, we randomly sample from W6A6 quantized diffusion models, and Figure 4 displays the generated images. These generated images demonstrate competitive performance and closely resemble real-world pictures.

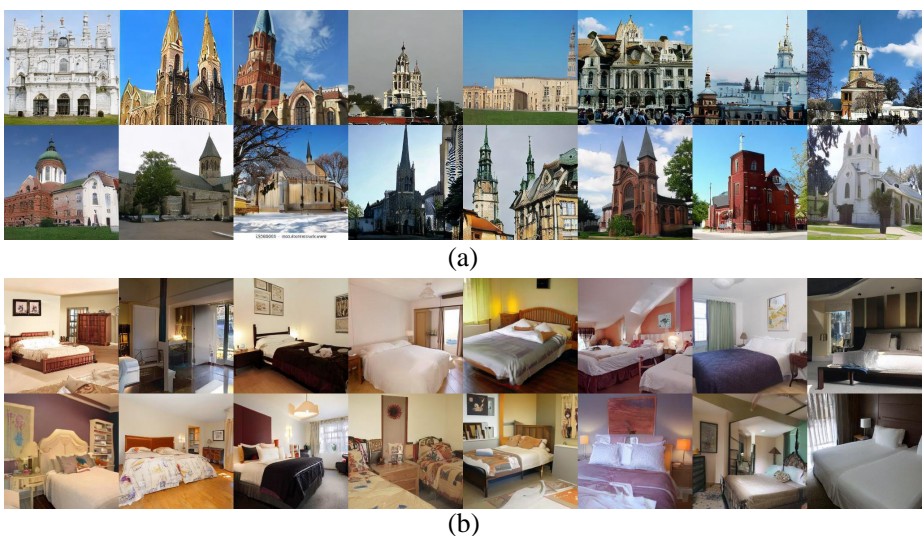

(a)

(b)

Figure 4: (a) contains samples from W6A6 quantized LDM-8 model on LSUN-Churches 256 × 256. (b) contains samples from W6A6 quantized LDM-4 model on LSUN-Bedrooms 256 × 256.

