# OpenReview forum: "Temporal Distribution-aware Quantization for Diffusion Models"
_ICLR.cc/2025/Conference — Submitted to ICLR 2025_

### Official Review · Reviewer_4JD3 · 2024-10-29

**Soundness:** 3
**Presentation:** 2
**Contribution:** 3
**Rating:** 6
**Confidence:** 4

**Summary:**

This paper introduces a framework for Distribution-aware Quantization for Diffusion Models which sets out to solve the problem of the uneven distribution of quantization-based error throughout the network and the accumulation of external quantization errors. The proposed work does so by introducing a time-varying quantization scheme that dynamically adapts to the distribution of activation values, and by using a full-precision noise estimation network in order to measure and mitigate the quantization-induced errors and align the quantizers with the varying input distributions.

**Strengths:**

+ The paper is presenting a comprehensive set of actions in order to improve the quality of Distribution-aware Quantization (DAQ) in Diffusion Models.
+ The proposed ideas are inspired by practical shortcomings of existing DAQ methods.
+ The paper is presenting extensive experimental results that showcase the merit of the proposed ideas.

**Weaknesses:**

- The presentation of the paper can be improved. Currently the narrative is very verbose and dense, hence the proposed contributions and key ideas seem to be hidden under the text. Introducing a few descriptive figures would certainly help declutter the paper and guide the reader through the main contributions.

**Questions:**

(1) I would recommend the introduction of descriptive figures and the decluttering of the existing text in order to further and in a more easily digestible manner articulate the main contributions of the work.

---

> ### Author Response · Authors · 2024-11-17
>
> Thank you for the suggestion, we will revise it as soon as possible.

---

> > ### Comment · Reviewer_4JD3 · 2024-11-25
> >
> > Thank you, I acknowledge your response.

---

### Official Review · Reviewer_VC9H · 2024-11-02

**Soundness:** 2
**Presentation:** 3
**Contribution:** 2
**Rating:** 5
**Confidence:** 3

**Summary:**

This paper presents a new quantization method for diffusion models called Time Distribution-Aware Quantization (DAQ). The aim is to address the high costs of diffusion models in terms of storage and inference time. By analyzing the deficiencies of existing quantization methods on diffusion models, such as the accumulation of quantization errors caused by not considering the distribution changes of different layers and modules at different time steps, this paper proposes a distribution-aware fine-tuning framework to dynamically suppress the accumulation of quantization errors during the calibration process and uses a full-precision noise estimation network to optimize the quantized noise estimation network to better adapt to the input distributions at different time steps. Experiments are conducted on multiple benchmark datasets, and the results show that DAQ outperforms existing methods in performance and can be applied as a plug-in to existing PTQ models to enhance overall performance.

**Strengths:**

1.The article presents a novel DAQ method that fully considers the distribution changes of diffusion models at different time steps, effectively addressing the problem of quantization error accumulation. The developed distribution-aware fine-tuning framework and full-precision noise estimation network optimization method provide new ideas and technical means for quantizing diffusion models.

2.The article conducts comprehensive experimental evaluations on multiple widely used public benchmark datasets, covering different model architectures, enhancing the persuasiveness of the experimental results. By comparing with various existing advanced methods, the advantages of the DAQ method are clearly demonstrated.

3.The method proposed in the article can be applied as a plug-in to existing PTQ models, with good scalability and compatibility, and can significantly improve the performance of existing models. At the same time, the fine-tuning method of quantization factors and the cumulative error suppression strategy alleviate the key problems in the quantization process of diffusion models to a certain extent, helping to better deploy diffusion models on resource-constrained devices.

**Weaknesses:**

1.The quantization reconstruction granularity division strategy adopted in the article leads to uneven distribution of quantization errors within the network, and further research on internal quantization granularity is needed to further optimize the method.

2.In the experiments, the performance of the method on different hardware platforms is not evaluated in detail. In practical applications, hardware differences may have an impact on the acceleration effect and resource utilization efficiency of the model. Moreover, the experiments may still not cover all possible application scenarios and model variants. For some special cases or diffusion models in specific fields, the effectiveness and adaptability of the method need to be further verified.

3.The theoretical explanations of the distribution-aware fine-tuning framework and the full-precision noise estimation network optimization method are relatively intuitive, lacking deeper theoretical analysis. I think the analysis of the relationship between quantization error accumulation and model performance can be further deepened to better understand the internal mechanism and potential improvement directions of the method.

**Questions:**

1. Are there more in - depth theoretical analyses to support that the fine - tuning methods for quantization factors and the cumulative error suppression strategies can effectively alleviate quantization error problems under different data distributions and model architectures?

2. How does the proposed method ensure that the quantization process will not introduce new and unconsidered error sources when dealing with complex diffusion model structures?

3. Can the specific implementation details of some key concepts and operations in the method (such as the distribution - aware dynamic fine - tuning framework) be further clarified to reduce the ambiguity of expression?

4. Can more content combined with theoretical analysis be added to the discussion part of the experimental results to better explain the principles behind the experimental data?

5. Compared with other existing model quantization methods, how can the advantages of the DAQ method be more specifically manifested and quantified in practical application scenarios (especially in resource - constrained devices)?

6. How to further expand the application range of this method so that it is not limited to the image generation field currently experimented with? Can some possible expansion directions be proposed and preliminarily explored?

---

> ### Author Response · Authors · 2024-11-17
>
> 1. There is an imbalance in the calibration extent exists across different quantized modules. We address this issue by assigning different calibration iterations to individual quantized modules and fine-tuning them to minimize the differences between full-precision and quantized modules. As shown in Figure 2, the quantization errors are significantly alleviated.
> 2. Since the proposed method does not modify the architecture of diffusion models, we only need to calibrate the quantizers to reduce errors caused by post-training quantization.
> 3. We dynamically adjust the calibration iterations for each quantized module and select quantizers across the noise estimation network for fine-tuning based on the output distribution of each module.
> 4. Implemented on Q-Diffusion[1], our method consistently demonstrates superior performance in experimental evaluations. This is due to our reduction of the imbalance in calibration extent across different quantized modules, a factor that is not considered by other methods, leading to overfitting in some smaller modules when calibrated on limited calibration datasets.
> 5. The proposed method can be implemented efficiently and is compatible with other quantization approaches. Additionally, its time and GPU resource requirements are comparable to those of simple PTQ methods.
> 6. While imbalances in calibration extent are generally present across quantized modules, this issue is less pronounced in small models where module differences are minimal. However, in larger and more complex models, quantization errors become more significant. Therefore, we believe that a well-designed strategy for different quantized modules is crucial, particularly in large models.
>
> References [1] Q-Diffusion: Quantizing Diffusion Models

---

### Official Review · Reviewer_mDZ9 · 2024-11-03

**Soundness:** 2
**Presentation:** 2
**Contribution:** 2
**Rating:** 3
**Confidence:** 4

**Summary:**

This paper highlights that most existing PTQ methods for diffusion models overlook the accumulating quantization errors caused by distribution shifts across layers and blocks at different timesteps. To address this issue, it proposes a temporal distribution-aware quantization(DAQ) method. DAQ uses a distribution-aware finetuning framework to dynamically suppress quantization errors during calibration, then optimizes the quantized noise estimation network at each sampling step using a full-precision noise estimation network to better align with varying input distributions.

**Strengths:**

1.	It proposes DAQ method for diffusion models to reduce the accumulating quantization errors arising from the substantial distribution variations across distinct layers and blocks at different timesteps.
2.	It designs a distribution-aware finetuning framework to suppress accumulating quantization errors and use a full-precision noise estimation network to better align quantizers with varying input distributions.
3.	Extensive benchmark experiments demonstrate the effectiveness of the proposed method compared to state-of-the-art approaches.

**Weaknesses:**

1. The paper is poorly organized and contains multiple errors. For example, Figure 2’s image and caption do not match, with “Ours” showing larger errors than expected; line 55 has a citation error; the related work section is overly verbose, spanning two full pages, etc.
2. Experimental results are not very convincing:
- APQ-DM and TFMQ-DM outperform “Ours” on several datasets.
- In Table 1, FID and IS metrics are used for CIFAR-10 and ImageNet, yet Table 2 only reports FID, though including IS would be straightforward.
- Table 3 demonstrates the plug-and-play aspect only on CIFAR-10, a relatively simple dataset, which lacks impact.
- DAQ is presented as a finetuning method, yet the comparisons are only with non-finetuning methods, omitting direct comparisons with other finetuning-based quantization methods like EfficientDM and Quest.
3. Given that the proposed method requires fine-tuning, what is its GPU time-consuming?

**Questions:**

Please refer to the weaknesses part.

---

> ### Author Response · Authors · 2024-11-17
>
> 1. Regarding IS[1] and FID[2] scores, we implemented our methods based on Q-Diffusion[3], consistently achieving better performance compared to Q-Diffusion[3]. However, variations in implementation details and calculation methods for IS[1] and FID[2] scores can lead to significant differences. Additionally, we attempted to reproduce other quantization methods while incorporating our modules, as shown in Tables 3 and 5. The experimental results indicate that the methods work, although some issues arose during the reproduction of the baselines.
> 2. IS[1] is used to evaluate generation quality based on classification performance. While CIFAR-10[4] and ImageNet[5] datasets contain multiple classes, LSUN-Churches[6] and LSUN-Bedrooms[6] datasets only contain a single class. Therefore, we report only the FID[2] scores for LSUN-Churches[6] and LSUN-Bedrooms[6].
> 3. There are some quantization methods whose complexity and resource consumption increase with those of the full-precision model. However the complexity and resource consumption of our methods only depend on the structure of model. So we choose to compare DAQ with PTQ series methods.
> 4. Although the proposed method requires fine-tuning, its GPU time consumption is comparable to traditional PTQ methods, such as Q-Diffusion[1]. This is because we fine-tune only certain parameters within the quantizers while freezing the weights, ensuring that the additional time cost is negligible.
>
> References [1] Improved Techniques for Training GANs [2] GANs Trained by a Two Time-Scale Update Rule Converge to a Local Nash Equilibrium [3] Q-Diffusion: Quantizing Diffusion Models [4] Learning multiple layers of features from tiny images [5] ImageNet: A large-scale hierarchical image database [6] LSUN: Construction of a Large-scale Image Dataset using Deep Learning with Humans in the Loop

---

> > ### Comment · Reviewer_mDZ9 · 2024-12-03
> > **Thanks for your rebuttal**
> >
> > Thank you for the rebuttal. Some of my questions have been addressed. However, some issues from the first round remain unresolved, such as the competitiveness of APQ-DM and TFMQ-DM across many datasets. Additionally, the reviewer still believes that more evaluation metrics are important, as IS was evaluated on the LSUN dataset in reference [1]. For these reasons, I am maintaining my score.
> >
> > [1] EfficientDM: Efficient Quantization-Aware Fine-Tuning of Low-Bit Diffusion Models

---

> > > ### Author Response · Authors · 2024-12-03
> > >
> > > As reproducing the results reported for methods such as TFMQ-DM is challenging, we conducted only a subset of experiments for comparison (as shown in Tables 3 and 5). The IS and FID scores for other methods listed in Tables 1 and 2 are sourced directly from their respective publications. Additionally, while we calculated the IS score for LSUN datasets (approximately 2.7 ± 0.2), we did not post it in the paper, as this metric is meaningless for single-class generation task.

---

### Official Review · Reviewer_c5vr · 2024-11-04

**Soundness:** 2
**Presentation:** 2
**Contribution:** 2
**Rating:** 5
**Confidence:** 4

**Summary:**

Diffusion models often require tremendous storage overhead and time cost during inference due to the large scale of networks and multiple sampling steps. This paper studied post-training quantization to speed up the inference process, and proposed a distribution-aware finetuning framework that can reduce accumulation quantization errors across modules and sampling timesteps. The proposed method can also be applied to existing post-training quantization models to improve the performance. Quantitative and qualitative experiment results were provided to demonstrate the effectiveness.

**Strengths:**

1. The paper pointed out that quantization errors vary across different modules of the network by providing motivational experiment results. This showed the necessity of distribution-aware methods to reduce the errors in a clear way.

2. The paper proposed to calibrate the quantization error across sampling time steps, which may be of interest in further quantization studies for diffusion models.

**Weaknesses:**

1. The title and introduction of the paper suggest that the paper considers temporal behavior of quantization models. However, it is not formally established how the proposed method deals with quantization errors over different sampling timesteps. Only high-level description is provided in Section 3.2.2. Also, the temporal finetuning parameter F_t introduced in Section 3.2.1 is not clearly related to the timestep t. It is not specified whether F_t is a hyper-parameter for individual t or a parameterized function of t. The lack of formal formulation weakens the significance of the proposed approach.

2. The performance of the proposed method is mixed as shown in experiments. The IS and FID scores in Tables 1 and 2 are not consistently better than previous models.

**Questions:**

1. In Section 3.1.1 the paper mentioned that the results in Figure 1 indicate that the number of iterations required for quantization reconstruction varies among different modules. But doesn’t Figure 1 show that the errors of all modules reduce and converge as the number of iterations increases?

2. In Section 3.2.1, how is F_t defined in terms of t? If F_t is a hyper-parameter for each t, is it only applicable to a fixed sequence of sampling timesteps?

---

> ### Author Response · Authors · 2024-11-17
>
> 1. The errors of all modules decrease and converge as the number of iterations increases. However, previous studies share a uniform setting for the number of iterations during the calibration process, regardless of the quantization granularity (modules). While some modules consist of a single layer, others may include more than five layers. Due to the varying complexity of these modules, their convergence speeds differ, indicating that each module requires its own specific number of iterations.
> 2. We initially set F_t=1 and then adjust it during the calibrating and fine-tuning processes. This learnable parameter is applicable to any sequence of sampling timesteps.
> 3. Regarding IS[1] and FID[2] scores, we implemented our methods based on Q-Diffusion[3], consistently achieving better performance compared to Q-Diffusion[3]. However, variations in implementation details and calculation methods for IS[1] and FID[2] scores can lead to significant differences. Additionally, we attempted to reproduce other quantization methods while incorporating our modules, as shown in Tables 3 and 5. The experimental results indicate that the methods work, although some issues arose during the reproduction of the baselines.
>
> References [1] Improved Techniques for Training GANs [2] GANs Trained by a Two Time-Scale Update Rule Converge to a Local Nash Equilibrium [3] Q-Diffusion: Quantizing Diffusion Models

---

### Official Review · Reviewer_jhNV · 2024-11-10

**Soundness:** 2
**Presentation:** 2
**Contribution:** 1
**Rating:** 3
**Confidence:** 5

**Summary:**

This paper proposes a framework dubbed DAQ for diffuison model quantization, which consists of a distribution-aware finetuning framework and a knowledge distillation strategy. The paper validates the proposed method on Cifar dataset and LSUN datasets.

**Strengths:**

1. This paper compares the proposed method to various baseline methods, and shows the proposed method can be used as a plug-and-play.
2. The proposed method is straightforward and might be easy to follow.

**Weaknesses:**

1. The reviewer believes the experiments are not sufficient.

(a). The datasets and models used in the paper are too old. The method is only validated on Cifar, LSUN, and imagenet with traditional small unet-based models, which is less meaningful. It would be better to do experiments on advanced text-to-image models, at least SD series. Besides, considering that DiT-based models have predominated, it is also necessary to experiment on DiT-based diffusion models such as Pixart, FLUX, etc.

(b). One key compoent of the proposed method is the timestep-aware strategy, however, the used sampling strategy (DDIM ) used in the paper is too old, which requires 50-100 steps. However, the reviwer is not sure if the proposed method can still be effective in few-steps scenarios. So, it would be better to validate the proposed method on few-steps diffusion models, such as LCM, SDXL-turbo, and FLUX, which only require 1-8 sampling steps.

2. Limited novelty. The ideas of the proposed method seem to be well explored in previous literature.

(a).  The paper proposes to minimize the difference between the full-precision model and the the quantized model, which is actually a common knowledge distillation method and has been proposed by EfficientDM[1].

(b) The proposed parameter-finetuning framework is just a combination of LSQ+[2] (which proposes to finetune both scaling factors and zero points) and the strategy of setting different quantizers for different steps (which is adopted by EfficientDM[1], TDQ[3] and APQ-DM[4] ).



References
[1] Efficientdm: Efficient quantization-aware fine-tuning of low-bit diffusion models
[2] LSQ+: Improving low-bit quantization through learnable offsets and better
Initialization
[3] Temporal Dynamic Quantization for Diffusion Models
[4] Towards accurate data-free quantization for diffusion models

**Questions:**

1. The reviewer is curious about Figure 1. Figure 1 shows different layers may have different optimal iterations for reconstuction.  Does the proposed method use different iterations for different layer? And how to determine the optimal iterations for each layer?

---

> ### Author Response · Authors · 2024-11-17
>
> 1. For the problem illustrated in Figure 1, we employ different numbers of iterations for each layer. The optimal number of iterations is determined dynamically by evaluating the output of the quantized layer in comparison with its full-precision counterpart.
> 2. Regarding the experiments, we implement our methods on several UNet-based models, following prior quantization approaches for diffusion models. This implementation demonstrates the effectiveness of our module as a plugin for frameworks such as Q-Diffusion[1], PTQD[2], EfficientDM[3], and TDQ[4].
> 3. In terms of novelty, we dynamically adjust the calibration iterations for each quantized module and selectively fine-tune quantizers across the noise estimation network based on the output distribution of different modules.
>
> References [1] Q-Diffusion: Quantizing Diffusion Models [2] PTQD: Accurate Post-Training Quantization for Diffusion Models [3] EfficientDM: Efficient Quantization-Aware Fine-Tuning of Low-Bit Diffusion Models [4] Temporal Dynamic Quantization for Diffusion Models

---

### Comment · Area_Chair_Zxgh · 2024-12-02

Dear reviewers,

Could you please help to take a look at the author responses? Thank you very much!

Best regards,

AC

---

### Meta-Review · Area_Chair_Zxgh · 2024-12-21

**Metareview:**

This paper presents DAQ, a Distribution-Aware Quantization framework for diffusion models. The framework aims to address quantization errors due to distribution shifts across layers and timesteps by introducing a temporal fine-tuning mechanism and leveraging a full-precision noise estimation network. While the paper explores an important problem and provides some promising directions, it suffers from significant weaknesses, as highlighted by multiple reviewers. Several components of the proposed framework, such as timestep-aware quantization and knowledge distillation, are extensions or combinations of existing techniques (e.g., EfficientDM, TDQ, APQ-DM). Reviewers also raised issues about the experimental evaluation and the presentation. For these limitations, I would like to recommend rejecting this paper.

**Additional Comments On Reviewer Discussion:**

During the rebuttal period, 2 out of 5 reviewers responded to the authors' replies. In particular, Reviewer mDZ9 mentioned that some issues remain unresolved and thus he/she did not change the score.

---

### Decision · Program_Chairs · 2025-01-22

Reject